# LATENT VISUAL REASONING

**Bangzheng Li**[1][*], **Ximeng Sun**[2], **Jiang Liu**[2], **Ze Wang**[2], **Jialian Wu**[2],
**Xiaodong Yu**[2], **Hao Chen**[2], **Emad Barsoum**[2], **Muhao Chen**[1], **Zicheng Liu**[2]
[1]University of California, Davis    [2]Advanced Micro Devices, Inc.
✉bzhli@ucdavis.edu

🌐 Website    ○ Code    🤗 Model

## ABSTRACT

Multimodal large language models (MLLMs) have achieved notable gains in various tasks by incorporating chain-of-thought (CoT) reasoning in language spaces. Recent work extends this direction by leveraging external tools for visual editing, thereby enhancing the visual signal along the reasoning trajectories. Nevertheless, these approaches remain fundamentally constrained: reasoning is still confined to the language space, with visual information treated as static preconditions. We introduce Latent Visual Reasoning (**LVR**), a new paradigm that enables autoregressive reasoning directly in the visual embedding space. A visual encoder first projects images into visual tokens within a joint semantic space shared with the language model. The language model is then trained to generate latent states that reconstruct key visual tokens critical for answering the query, constituting the process of latent visual reasoning. By interleaving **LVR** with standard text generation, our model achieves substantial gains on perception-intensive visual question answering tasks. In addition, we adapt the GRPO algorithm to conduct reinforcement learning on latent reasoning, further balancing **LVR** and textual generation. We show that **LVR** substantially improves fine-grained visual understanding and perception, achieving 71.67% on MMVP compared to 66.67% with Qwen2.5-VL. The Code base and model weights will be released later.

## 1 INTRODUCTION

Multimodal large language models (MLLMs) (Li et al., 2024b; Bai et al., 2025b; Wang et al., 2025d) have shown remarkable capability in jointly understanding visual and textual content. By leveraging the generative capabilities of their backbone large language models (LLMs), MLLMs extend the expressiveness of visual encoders beyond simple perception tasks. This advancement has enabled the integration of chain-of-thought (CoT) reasoning into MLLMs, allowing them to perform structured textual reasoning in response to complex multimodal queries. In this paradigm, the LLM decomposes a query into intermediate steps and resolves each step while conditioning on static visual inputs. This approach, referred to as "Thinking about Images" by (Su et al., 2025d), has proven effective across diverse domains, including scientific visual question answering (Zhang et al., 2023), mathematics (Huang et al., 2025a), and visual grounding (Bai et al., 2025c).

Further research has expanded the multimodal reasoning workspace by enabling active editing of input images alongside textual reasoning trajectories. Such editing includes drawing auxiliary lines (Hu et al., 2024), zooming in (Shao et al., 2024a; Su et al., 2025a; Surís et al., 2023), shifting image styles (Liu et al., 2025a), highlighting sub-regions (Fu et al., 2025), and more. During intermediate CoT steps, methods in this paradigm either call external tools or generate programs to manipulate images, re-encode the edited outputs, and inject the new image tokens as visual enhancements into subsequent textual reasoning. In this way, the salient visual information is actively incorporated throughout the reasoning process. These methods are commonly termed "Thinking with images."

---

[*]Work done during an internship at AMD.

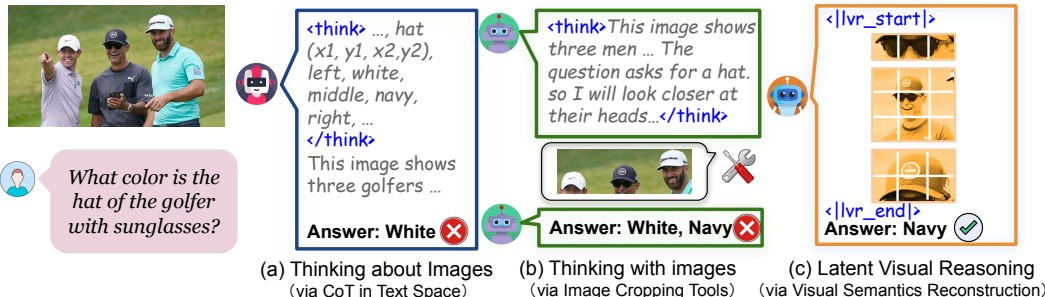

Figure 1: **Conceptual illustration of LATENT VISUAL REASONING (LVR).** We compare **LVR** with two paradigms: "Think about images," which performs multimodal reasoning entirely in text space, and "Think with images", which leverages external visual tools to highlight regions of interest (ROIs). In contrast, **LVR** leverages the LLM's latent space to reconstruct the semantics of ROIs, enabling **seamless cross-modal reasoning.**

Both "Thinking about Images" and "Thinking with Images" aim to address a core limitation in current MLLMs: despite sophisticated visual encoders projecting visual information into text spaces, backbone LLMs often fail to capture the visual details most relevant to the text query. This shortcoming arises from factors such as modality projection bias (Zhang et al., 2024; Liu et al., 2023), modality interference (Cai et al., 2025; Wang et al., 2025e; Pezeshkpour et al., 2025; Deng et al., 2025a), and cross-modality attention bias (Zhang et al., 2025c). "Thinking about Images" addresses these issues by generating additional task-relevant text tokens in the context window, thereby increasing the likelihood of correct answers. However, excessive token generation may cause the textual context to dominate, overshadowing essential visual inputs (Huang et al., 2024). In contrast, "Thinking with Images" leverages external tools to inject visual information during text generation, calibrating the alignment between generated text and the original visual input. Yet these approaches often bypass the newly injected sub-images due to training data bias, or remain constrained by the predefined operations of the tools (Su et al., 2025a). In short, both categories primarily refine text generation to improve cross-modality understanding, yet a fundamental gap persists between visual inputs and text generation in producing the final answer.

A parallel line of research explores omni-modality foundation models that accept both text and visual inputs and generate outputs in both modalities simultaneously (Team, 2024; Deng et al., 2025b; Xie et al., 2025). Some subsequent works attempt to exploit image generation capabilities for multimodal reasoning, but effectiveness has so far been demonstrated only in specific downstream tasks such as navigation and maze solving (Li et al., 2025; Xu et al., 2025). Moreover, it remains unclear whether visual inputs, once decoded and re-encoded by such models, can still faithfully preserve the original information. Rethinking the input of MLLMs, where continuous visual tokens and discrete text tokens are embedded in a shared latent semantic space, we ask the following question:

> *If visual and textual tokens are embedded in a joint semantic space within an MLLM, why not reason over both jointly as well?*

It is both natural and efficient to extend reasoning beyond discrete text tokens to include visual tokens that directly encode visual information. However, conventional LLMs are limited to operating on discrete tokens due to their next-token prediction training objective. To address this, Hao et al. proposed passing last hidden states rather than text tokens, enabling more efficient expression of complex thoughts through latent reasoning. Building on this idea, we introduce LATENT VISUAL REASONING (**LVR**), a novel paradigm for multimodal reasoning (see Fig. 1). Our approach involves a simple yet fundamental modification to the conventional Vision–Projector–LLM structure: the LLM is able to perform hybrid reasoning that alternates between LVR and standard text generation. In the LVR phase, the LLM leverages the last hidden state to approximate the question-relevant visual tokens within the visual inputs. During the text generation phase, the model predicts the next text token in sequence. Both phases operate in an auto-regressive manner.

We propose a two-stage training pipeline for **LVR**. The first stage is supervised fine-tuning (SFT), which jointly optimizes **LVR**'s internal processes alongside next-token prediction for text generation. The second stage applies reinforcement learning (RL), allowing **LVR** to self-evolve the latent

reasoning process while receiving policy rewards from generated text, thereby encouraging a more unified semantic space. Specifically, we adapt the GRPO algorithm (Shao et al., 2024c) to replay latent reasoning steps during policy gradient loss computation. In addition, GRPO leverages verifiable rewards to evaluate roll-out responses, where the policy gradient loss is computed solely from the token distribution of the text generation component. Experimental results demonstrate that **LVR** achieves substantial improvements over state-of-the-art MLLMs, particularly on perception-intensive and visual detail-dependent understanding tasks.

In summary, our contributions are as follows:

- We propose LATENT VISUAL REASONING, a novel multimodal reasoning paradigm that unifies latent reasoning over visual inputs with text generation in the language space, enabling deeper integration of visual and textual signals throughout the model's reasoning process.

- We introduce architectural innovations and training frameworks for stable and scalable training MLLMs with **LVR**. Our approach combines a reconstruction loss with next-token prediction for SFT and extends the GRPO algorithm to latent reasoning for reinforcement learning.

- Through extensive evaluation, we demonstrate that **LVR** achieves strong performance across diverse visual question answering benchmarks requiring fine-grained visual understanding and perception. In addition, our comprehensive ablation studies and discussions explore alternative architectural designs and training objectives, providing insights to guide future research on this emerging paradigm.

## 2 RELATED WORKS

**Think about Images.** Many prior works have employed text-space chain-of-thought (CoT) reasoning to enhance visual perception and multimodal mathematical reasoning. Early approaches focused on constructing SFT datasets (Xu et al., 2024; Shao et al., 2024a; Wei et al., 2025b), aiming for models to fully acquire such reasoning patterns during training. More recently, the field has shifted toward RL–based methods (Peng et al., 2025; Tan et al., 2025; Meng et al., 2025; Yang et al., 2025a) with many discussions on data design (Liang et al., 2025), loss design (Hong et al., 2025) and training stages (Wei et al., 2025a; Deng et al., 2025c; Liu et al., 2025b; Chen et al., 2025b). Some studies explore specialized RL phase designs, while others mitigate visual hallucination by generating auxiliary captions (Xia et al., 2025) or randomly masking parts of the image (Wang et al., 2025g). Additional efforts guide models to focus on regions of interest (ROIs) by predicting points, bounding boxes, or descriptions (Jiang et al., 2025; Ni et al., 2025; Yu et al., 2025; Liu et al., 2025c), ensuring that answers are grounded in the correct visual evidence. Another line of work incorporates verification or rewriting steps to refine reasoning quality (Wang et al., 2025c; Shen et al., 2025a; Zhang et al., 2025b; Chen et al., 2025a). Despite these advances, most methods still perform reasoning in text space, which remains an indirect and inefficient representation of visual understanding. Humans, by contrast, can reason about images naturally without translating them into text. Inspired by this observation, our work seeks to more closely mimic human visual reasoning by enabling models to understand and reason directly in the visual space.

**Think with Images.** Another recent line of research emphasizes augmenting multimodal models with external, predefined visual tools. Many approaches employ zoom-in or cropping utilities (Su et al., 2025b; Zhang et al., 2025d) to locate ROIs relevant to a given question, while others integrate more advanced tools such as OCR engines, chart parsers, or even drawing interfaces (Huang et al., 2025b). To determine when and how to invoke these tools, early studies relied on supervised fine-tuning (Wang et al., 2025b; Zhang et al., 2025a; Chung et al., 2025), whereas more recent work adopts reinforcement learning to learn tool-using behaviors (Zhang et al., 2025d; Su et al., 2025c; Geng et al., 2025; Wu et al., 2025a), enabling interleaved CoT reasoning and tool execution (Wu et al., 2025b; Zheng et al., 2025). Despite their success, these approaches remain constrained by the availability and design of external tools. Tool APIs can be difficult to extend, and updates or changes often require substantial training effort. Moreover, many fundamental operations, such as zooming, cropping, or OCR, can potentially be solved directly within modern MLLMs without tool invocation. Motivated by these limitations, our work explores latent visual reasoning, approximating question-relevant visual tokens directly in the visual representation space rather than relying on explicit external tools.

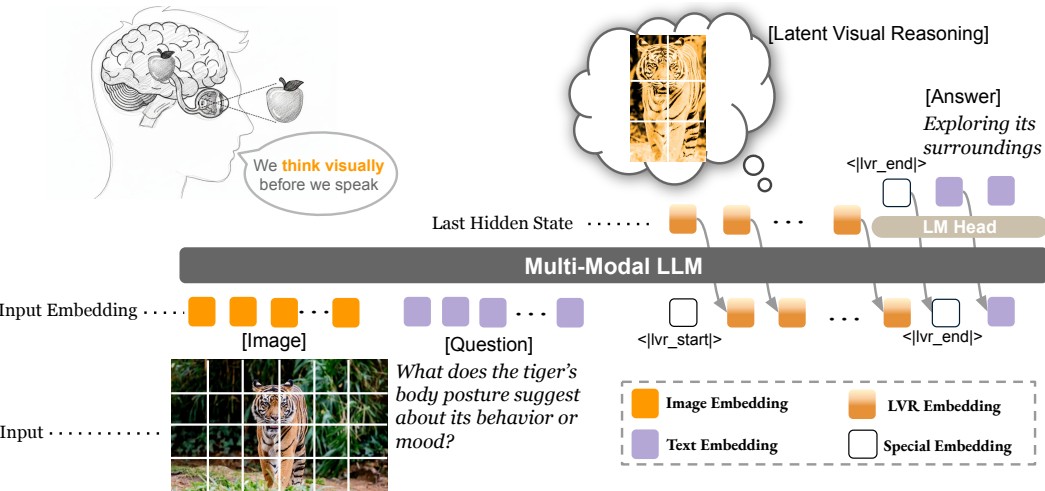

Figure 2: **Training and inference pipeline of LVR.** The overall framework closely follows a standard MLLM. Images are encoded into tokens by a visual encoder and mapped into a joint semantic space with text embeddings. During the SFT stage, bounding boxes are provided to identify query-relevant visual tokens, which supervise the last hidden states in the **LVR** process. Here, only the LLM's last hidden states are passed forward for latent reasoning, optimized with a mean squared error (MSE) loss. The **LVR** process is wrapped with special tokens that indicate reasoning mode. Once all query-relevant visual tokens are consumed, the model exits **LVR** and resumes standard text generation with cross-entropy loss. During RL training, the model self-evolves the **LVR** process learned in SFT, while only the text generation part is supervised, using our adapted $GRPO_{latent}$. At inference, the model triggers **LVR** upon generating the special token, propagates hidden states to reconstruct visual semantics, and resumes text generation when a stopping criterion is met.

**Latent Reasoning.** In natural language processing, several studies have explored performing reasoning in the latent space (e.g., the final layer outputs) rather than directly in the token space (Hao et al., 2024). Some work (Shen et al., 2025b) leverages latent representations to approximate token-level reasoning, investigating both fixed-length and variable-length latent reasoning strategies (Cheng & Van Durme, 2024). However, latent spaces in NLP are often difficult to interpret, making supervision of such representations challenging. Building on this idea, we extend latent-space reasoning to the visual domain, where the latent tokens are grounded in visual meaning. Specifically, we aim for these tokens to approximate question-relevant visual features. Recent efforts have also explored similar directions, using latent spaces to capture visual content, but many rely on auxiliary images to supervise latent tokens (Yang et al., 2025b; Bigverdi et al., 2025). Such auxiliary data introduces additional labeling and pairing costs, ultimately limiting scalability. In contrast, our approach requires no extra images and can be readily applied across diverse vision tasks after a single training.

## 3 LATENT VISUAL REASONING

In this section, we introduce LATENT VISUAL REASONING, a new paradigm that enables MLLMs to reason jointly over text and visual tokens. The overall inference pipeline is illustrated in Figure 2. At a high level, **LVR** is trained to reconstruct visual semantics relevant to both the input image and the accompanying text query. These reconstructed semantics, which we term *latent visual thoughts*, are then combined with the original inputs to guide the generation of textual responses.

We begin by providing an overview of the **LVR** architecture, which is built on the Qwen-2.5-VL series (Bai et al., 2025a) (§3.1). Next, we present a two-stage training pipeline—combining supervised fine-tuning and reinforcement learning—that jointly teaches the model to reconstruct visual semantics and generate text (§3.2). Finally, we introduce decoding strategies that allow the model to flexibly alternate between LATENT VISUAL REASONING and standard text generation (§3.3) during the inference.

### 3.1 METHOD OVERVIEW

Fig. 2 illustrates the architecture of **LVR**, which largely follows the standard MLLM design. It consists of three key components: a vision encoder $vision(\cdot)$, an LLM backbone $\theta(\cdot)$, and a multimodal projector $proj(\cdot)$ that aligns the two modalities. Given an input image–question pair $(\mathbf{X}_v, \mathbf{X}_t)$, the vision encoder transforms the image into visual features $\mathbf{V} = vision(\mathbf{X}_v)$. In parallel, the textual question $\mathbf{X}_t$ is embedded into language features $\mathbf{T}$ by the LLM's embedding layers. Since visual and textual features reside in distinct latent spaces, the projector $proj(\cdot)$ maps the visual features into a representation aligned with the language model's latent space, denoted as $\mathbf{V}_T = proj(\mathbf{V})$.

In standard MLLMs, both $\mathbf{V}_T$ and $\mathbf{T}$ are passed into the LLM $\theta(\cdot)$. However, the decoding process remains strictly text-centric: hidden states are computed auto-regressively and mapped, through the language modeling head, to discrete tokens in the vocabulary. Thus, while visual information can guide the reasoning process, the output space is still constrained to text tokens, limiting the model's ability to directly reason over visual semantics.

To overcome this limitation, we propose **LVR**, which extends the conventional text-only generation paradigm by enabling interleaved LATENT VISUAL REASONING. When the special token $< |\texttt{lvr\_start}| >$ is generated, the model enters a latent reasoning mode, where it reconstructs visual semantics in the space of $\mathbf{V}_T$ to better support answering the textual query. The hidden states produced in this mode are directly propagated as input embeddings to subsequent positions until a stopping criterion is reached. At that point, the model generates the token $< |\texttt{lvr\_end}| >$ and resumes standard text generation. The hidden-state sequence produced during the **LVR** segment can be regarded as an analogue of a human's "visual thinking," where query-relevant visual semantics are mentally reconstructed to enhance the precision of reasoning and answering.

### 3.2 TWO-STAGE TRAINING PIPELINE

#### 3.2.1 SUPERVISED FINE-TUNING.

We begin with supervised fine-tuning (SFT) to instill the basic pattern of latent visual reasoning. During this stage, we explicitly supervise the reasoning content by forcing the MLLM to use its latent space embeddings (i.e., last hidden states) to reconstruct the ground-truth regions of interest for each image–text pair. Since the reconstruction loss directly dictates what the model must reason about, the reasoning process is constrained rather than free-form. We therefore characterize this stage as a teacher-forcing paradigm for latent visual reasoning: although restrictive, it allows the model to quickly acquire the fundamental ability to reason in the latent space.

Each SFT instance consists of an image-question pair, accompanied by a pre-annotated bounding box of a region of interest (ROI) relevant to the query. For a given image, the model's image processor first crops it into a grid of visual patches. Based on the ROI bounding box, **LVR** efficiently selects the corresponding patches and retrieves their indices $\mathbf{I} = \{I_1, I_2, I_3, \ldots, I_{T_v}\}$ from the sequence of flattened visual patches in $O(1)$ time. During the forward pass, the image is encoded into a sequence of visual embeddings in the semantic space, followed by the embeddings of the text tokens. A subset of visual embeddings $\mathbf{v} = \{v_1, v_2, \ldots, v_{T_v}\}$ is then gathered using the index list $\mathbf{I}$, which—by design of the ViT encoder—directly corresponds to the visual patches within the ROI. These selected tokens are enclosed by the special tokens $< |\texttt{lvr\_start}| >$ and $< |\texttt{lvr\_end}| >$, thereby defining the LATENT VISUAL REASONING process. Finally, the remaining textual response is appended to the sequence for generation.

We train our model with two joint learning objectives that explicitly couple latent visual reasoning with downstream text generation.

**Visual Reconstruction Loss.** During latent reasoning, the model predicts a sequence of the final hidden states $\{\mathbf{h}_t\}_{t=1}^{T_v}$ that are expected to encode the underlying visual semantics. We enforce these hidden states to approximate the ground-truth visual embeddings $\{\mathbf{v}_t\}_{t=1}^{T_v}$ via a mean squared error (MSE) objective:

$$\mathcal{L}_{\text{LVR}} = \frac{1}{T_v} \sum_{t=1}^{T_v} \left\| \mathbf{h}_t - \mathbf{v}_t \right\|_2^2. \tag{1}$$

**Next-Token Prediction (NTP) Loss.** For the subsequent language modeling phase, the model generates the final response tokens $\{y_t\}_{t=1}^{T_y}$ to answer the visual question. We adopt the standard cross-entropy (CE) loss to maximize the likelihood of the ground-truth sequence:

$$\mathcal{L}_{\text{NTP}} = -\frac{1}{T_y} \sum_{t=1}^{T_y} \log p_\theta(y_t \mid y_{<t}, \mathbf{h}_{1:T_v}). \tag{2}$$

**Joint Objective.** The overall training loss is a weighted sum of the two components:

$$\mathcal{L} = \mathcal{L}_{\text{NTP}} + \lambda_{\text{LVR}} \cdot \mathcal{L}_{\text{LVR}}, \tag{3}$$

where $\lambda_{\text{LVR}}$ is a hyperparameter to balance reconstruction and generation signals.

### 3.2.2 REINFORCEMENT LEARNING

We employ GRPO to further refine the interaction between **LVR** and standard text generation. In this stage, rewards are computed solely based on output accuracy and adherence to the required response format. Unlike supervised fine-tuning, no constraints are imposed on the intermediate outputs of **LVR**, allowing the model to freely explore the latent visual reasoning space. This relaxation also removes the need for pre-annotated ROI bounding boxes, as GRPO training can be performed directly on image–text pairs. Moreover, GRPO implicitly encourages the generation of the $< |\texttt{lvr\_start}| >$ token, thereby increasing the likelihood of activating the **LVR** process during response generation.

However, directly applying the standard GRPO algorithm to **LVR** is non-trivial. The key challenge lies in the mismatch between the policy-gradient loss, which is defined over token distributions, and the latent reasoning process, which lacks an explicit token distribution. **To address this issue, we propose $GRPO_{latent}$, a variant that can be seamlessly applied to any latent-reasoning model.**

$$J_{\text{GRPO}_{\text{latent}}}(\theta) = \mathbb{E}_{q,I,\, o\sim\pi_{\theta_{\text{old}}}} \left[ \frac{1}{|y|} \sum_{t=1}^{|y|} \min\left( r_t(\theta)\,\hat{A}_t,\, \text{clip}\big(r_t(\theta),\, 1-\varepsilon,\, 1+\varepsilon\big)\,\hat{A}_t \right) \right.$$
$$\left. - \beta\, D_{\text{KL}}\big(\pi_\theta(\cdot \mid q, I) \,\|\, \pi_{\text{ref}}(\cdot \mid q, I)\big) \right] \tag{4}$$

Here, the token-wise importance ratio $r_{i,t}(\theta)$ for a text token $y_{i,t}$ is computed using a teacher-forcing log-probability pass, where **the latent reasoning hidden states are replayed from the original rollout**:

$$r_{i,t}(\theta) = \frac{\pi_\theta(y_{i,t} \mid q, I, \widetilde{h}_i^{\text{latent}}, y_{i,<t})}{\pi_{\theta_{\text{old}}}(y_{i,t} \mid q, I, \widetilde{h}_i^{\text{latent}}, y_{i,<t})} \tag{5}$$

During rollout we record the last hidden states of **LVR** processes

$$\widetilde{h}_i^{\text{latent}} = \{h_{i,1}^{\text{latent}}, \dots, h_{i,L}^{\text{latent}}\}.$$

To evaluate importance ratios, we perform a teacher-forcing forward pass under both $\pi_\theta$ and $\pi_{\theta_{\text{old}}}$. The recorded hidden states $\widetilde{h}_i^{\text{latent}}$ are patched into the latent-reasoning positions of the model, thereby restoring the exact context preceding text generation and ensuring consistent conditional log-probabilities for the output sequence.

Rewards are derived solely from the text output $y$. We incorporate two reward types: a **format reward**, which equals 1 if the response contains both $< |\texttt{lvr\_start}| >$ and $< |\texttt{lvr\_end}| >$ tokens and 0 otherwise, and an **accuracy reward**, which equals 1 if the answer is correct and 0 otherwise. The format reward will encourage **LVR** process in the response while the accuracy reward indirectly supervises the latent reasoning process through its impact on text generation.

Finally, the group-normalized reward is defined as:

$$\tilde{R}_i = \frac{R(y_i) - \text{mean}(R(y_1), \dots, R(y_G))}{\text{std}(R(y_1), \dots, R(y_G))}, \qquad \hat{A}_{i,t} = \tilde{R}_i \quad (\forall t \in \{1, \dots, |y_i|\}). \tag{6}$$

## 3.3 DECODING STRATEGIES

Our initial results reveal that decoding in **LVR** can be challenging, as it is often unclear when the model should exit the **LVR** process. Specifically, when the next-token prediction introduces a $<$ $|\text{lvr\_start}| >$ token, the decoder switches into the **LVR** mode, passing the last hidden states instead of the language model head's predicted tokens. During this phase, the language model head continues producing token predictions, but it should ideally emit $< |\text{lvr\_end}| >$ once an optimal length of latent reasoning has been reached, reconstructing all necessary visual semantics for the current task. In practice, however, the predicted tokens during latent reasoning are often unstable, motivating us to propose three decoding strategies:

- **Fixed Token**: assigns a constant budget of reasoning steps. Once the budget is reached, the model immediately exits latent reasoning mode.
- **Latent End Token**: introduces a trainable tensor in the hidden state space. When the last hidden state approaches this tensor, the decoder resumes text-token generation.
- **Mode Switching Loss**: adds an auxiliary loss term during SFT that supervises the token distribution predicted by the language model head in the latent reasoning phase. A BCE loss encourages the distribution of the final latent reasoning token toward $< |\text{lvr\_end}| >$ (close to 1), while all intermediate tokens are pushed away from $< |\text{lvr\_end}| >$ (close to 0). At inference time, the model exits latent reasoning mode once $< |\text{lvr\_end}| >$ is predicted.

Empirically, we find that **Fixed Token** achieves the best performance, while **Mode Switching Loss** fails to work as intended. A detailed analysis is provided in Section 4.5.

## 4 EXPERIMENT

We adopt Qwen-2.5-VL 3B and 7B as the backbone MLLMs. For the visual encoder, we set the maximum resolution to $5120 \times 28 \times 28$ pixels and the minimum to $128 \times 28 \times 28$ pixels. In both training stages, the visual encoder and multimodal projector are kept frozen, with only the LLM parameters updated. This design reflects the learning objective of **LVR** to unify the reasoning space under the hypothesis that an optimal modality projection can be achieved without additional tuning.

In the SFT stage, we use VISUAL COT (Shao et al., 2024b) as the primary training data. This large-scale VQA dataset contains 438k question–answer pairs, each annotated with bounding boxes that mark the critical regions needed to derive the answer. During training, the variable number of image tokens and **LVR** tokens results in imbalanced instance lengths. To address this, we adopt the adaptive multimodal data-packing strategy from Chen et al. (2024), which supports dynamic batching: multiple shorter instances can be packed together, while longer ones are grouped in smaller numbers. On average, this yields an effective batch size of $\sim$3.2 per device. The learning rate is set to $1 \times 10^{-5}$. For the 7B variant, supervised fine-tuning requires roughly 40 hours to complete 2,500 steps on a $4 \times$ AMD MI250 GPU cluster.

For reinforcement learning, we implement a latent GRPO framework customized from the Hugging-Face TRL package, using ViRL (Wang et al., 2025a) as the training data. The policy generates 8 responses per input, with a learning rate of $1 \times 10^{-5}$, a fixed sampling temperature of $\tau = 0.9$, and a KL coefficient of $\beta = 0.04$. This stage is applied only to the 3B variant, requiring about 20 hours to complete 1,500 steps. We do not extend it to the 7B variant due to limited computational resources.

### 4.1 EVALUATION BENCHMARKS

We evaluate **LVR** on the visual detail understanding task and a diverse set of vision-centric benchmarks. For visual detail understanding, we adopt $V^*$ Bench, which assesses MLLMs' ability to perform fine-grained visual detail search and relative spatial reasoning respectively on two subsets. We further employ MMVP (Tong et al., 2024) to measure perception robustness under subtle image perturbations, providing a rigorous test of **LVR** 's fine-grained reasoning capabilities.

Beyond these, we evaluate on Counting (object enumeration), JigSaw (image reconstruction from fragments), Relative Reflectance (pixel-level albedo comparison), and Spatial Relation (object–relation understanding within a scene). These tasks are drawn from BLINK (Fu et al., 2024), a benchmark of expert-annotated, perception-heavy tasks designed for MLLMs.

| Method | $V^*$ | $V^*_{D.A.}$ | $V^*_{R.P.}$ | MMVP | Counting | IQ-Test | JigSaw | Relative Reflect | Spatial Relation |
|---|---|---|---|---|---|---|---|---|---|
| *Close Source Models* | | | | | | | | | |
| GPT-4o | 62.8 | - | - | - | 51.7 | 30.0 | 58.0 | 38.8 | 76.9 |
| Gemini2.5-Pro | 79.2 | - | - | - | - | - | - | - | - |
| ARGUS-X3 | 78.5 | - | - | 45.5 | - | - | - | - | - |
| *Open Models based on Qwen2.5-VL-7B* | | | | | | | | | |
| Qwen2.5-VL | 78.5 | 81.7 | 73.7 | 66.7 | 66.7 | 26.0 | 52.0 | 38.8 | 87.4 |
| PAPO | 36.1 | 25.2 | 52.6 | 54.3 | 66.7 | 29.3 | 52.0 | 39.6 | 88.8 |
| Vision-R1 | 70.2 | 70.4 | 69.7 | 46.7 | 51.7 | 26.7 | 27.3 | **44.8** | 66.4 |
| PixelReasoner | 80.1 | 81.7 | 77.6 | 67.0 | 66.7 | 25.3 | 52.7 | 42.5 | 88.1 |
| SFT | 79.1 | 82.6 | 73.7 | 65.7 | 67.5 | 26.7 | 45.3 | 33.6 | 88.8 |
| **LVR** (4 Steps) | 81.2 | **84.4** | 76.3 | **72.0** | 69.2 | 28.7 | **52.7** | 42.5 | **89.5** |
| **LVR** (8 Steps) | **81.7** | **84.4** | 77.6 | 71.7 | 70.0 | **29.3** | 52.0 | 42.5 | 86.0 |
| **LVR** (16 Steps) | 80.6 | 81.7 | **79.0** | 71.7 | **70.8** | 27.3 | **52.7** | 41.8 | 87.4 |

Table 1: **Experimental results on vision-centric tasks. LVR** outperforms both "Think about Images" and "Think with Images" , highlighting the scalability of this new paradigm for MLLMs.

| Method | $V^*$ | $V^*_{D.A.}$ | $V^*_{R.P.}$ | MMVP | IQ-Test | JigSaw |
|---|---|---|---|---|---|---|
| PAPO | 31.94 | 22.61 | 46.05 | 50 | 31.33 | 46.67 |
| **LVR** (4 \| 8 \| 16) | 64.9 \| 65.5 \| 66.5 | 69.6 \| 71.3 \| 71.3 | **60.5** \| **60.5** \| 56.6 | 54.7 \| 56.0 \| 56.0 | 29.3 \| 30.7 \| 30.0 | **52.7** \| **52.7** \| 52.0 |
| **LVR** $_{RL}$ (4 \| 8 \| 16) | 65.5 \| **67.0** \| 66.5 | 69.6 \| **72.2** \| 71.3 | 59.2 \| 59.2 \| 59.2 | 55.3 \| 55.3 \| **58.0** | 30.7 \| **32.0** \| 30.0 | **52.7** \| **52.7** \| 50.7 |

Table 2: **RL results with 3B models.** $GRPO_{latent}$ further boosts performance, demonstrating the effectiveness of adapting RL for latent reasoning and enabling self-evolution.

To ensure consistency and fairness, all evaluations follow the standardized metrics provided by LMMs-Eval (Li et al., 2024a).

## 4.2 BASELINES

We compare **LVR** against state-of-the-art MLLM baselines, grouped into the following categories:

**Thinking about Images.** This category includes Vision-R1 (Huang et al., 2025a) and PAPO (Wang et al., 2025f), which use reinforcement learning with verifiable rewards to equip MLLMs with chain-of-thought reasoning. Vision-R1 follows a "think before answer" trajectory, while PAPO adds an implicit perception loss for image-grounded descriptions.

**Thinking with Images.** This category includes PixelReasoner (Su et al., 2025a) and Argus-X3 (Man et al., 2025). PixelReasoner employs image-editing tools to iteratively enhance the input image during reasoning. In contrast, Argus-X3 detects regions of interest via bounding boxes, extracts the corresponding visual tokens, and reinjects them into the MLLM input to strengthen perception. This makes Argus-X3 a strong reference point for comparison with **LVR**: whereas Argus-X3 depends on external tools for feature extraction, **LVR** directly learns to reconstruct visual semantics.

To isolate the effect of training data, we also include a baseline trained with standard supervised fine-tuning on the same dataset as **LVR**, denoted as SFT. For completeness, we include each baseline model's system prompt when it is available in the documentation.

## 4.3 MAIN RESULTS

The main evaluation results are summarized in Table 1. We report performance under the best decoding strategy, **Fixed Token**, with **LVR** steps set to [4, 8, 16]. Overall, **LVR** achieves state-of-the-art results across most benchmarks. Importantly, for a fair comparison, all open-source baselines are built on the same backbone MLLMs as **LVR**, highlighting the effectiveness of latent reasoning over existing "Think about images" or "Think with images" approaches.

The largest gains are observed on the $V^*$ and MMVP benchmarks, where **LVR** consistently achieves top performance across all step settings. In particular, it improves the base model by 2.7% on the $V^*_{D.A.}$ subset, which measures visual detail search, and by 5.3% on the $V^*_{R.P.}$ subset, which evaluates relative spatial reasoning. On MMVP, which perturbs subtle image details to assess perception robustness, **LVR** demonstrates strong performance by reconstructing target visual semantics, accurately identifying differences, and thereby improving accuracy. Moreover, it surpasses PixelReasoner in both categories, despite PixelReasoner relying on external tools to crop image sub-regions.

| Method | $V^*$ | $V^*_{D.A.}$ | $V^*_{R.P.}$ | MMVP | IQ-Test | JigSaw |
|---|---|---|---|---|---|---|
| **LVR** | 81.7 | 84.4 | 77.6 | 71.7 | 29.3 | 52.0 |
| **LVR** $_{LatentEnd}$ | 39.8 | 32.2 | 51.3 | 19.0 | 6.7 | 13.3 |
| **LVR** $_{MLPHead}$ | 74.4 | 76.5 | 71.1 | 69.7 | 23.3 | 50.0 |
| **LVR** $_{GLUHead}$ | 79.6 | 82.6 | 75.0 | 69.0 | 25.3 | 44.0 |

Table 3: Ablation studies on the 7B model show the standard approach performs best, indicating the LLM natively aligns visual and textual semantics without an extra head. However, the unstable latent end token suggests a need for future work on variable-length reasoning.

This underscores that **reconstructing visual semantics can be more effective than depending on external visual-editing tools (as in "Think with Images") for fine-grained visual understanding.** In addition, both PAPO and Vision-R1 exhibit degraded performance on $V^*$, suggesting that **textual-space CoT in MLLMs (i.e., "Think about Images") may introduce cross-modal interference and weaken perception, whereas LVR avoids this issue by reasoning jointly across modalities.**

**LVR** also achieves leading results on Counting, IQ-Test, JigSaw, and Spatial Relation. In Counting, the model is required to enumerate specific objects in a scene; in IQ-Test, it solves geometry-based puzzles; In JigSaw, it reassembles image crops; and in Spatial Relation, it answers questions about relative object positions. These results demonstrate that LATENT VISUAL REASONING effectively unifies object detection, visual-dependent logical reasoning, visual reconstruction, and spatial relation understanding.

However, **LVR** does not achieve top performance on Relative Reflect. We attribute this gap to a distribution shift between training and evaluation data: such tasks require reasoning over multiple images, whereas **LVR** was trained exclusively on single-image data. We anticipate that incorporating cross-image data augmentation in future work will further strengthen **LVR** 's capability for multi-image reasoning.

## 4.4 RL RESULTS

We present the results of our proposed $GRPO_{\text{latent}}$ in Table 2. Reinforcement learning further enhances **LVR** performance beyond the SFT stage across multiple benchmarks. The superior results demonstrate that incorporating a format reward based on the $< |\texttt{lvr\_start}| >$ and $< |\texttt{lvr\_end}| >$ tokens effectively encourages the model to perform LATENT VISUAL REASONING. In contrast, removing these trigger tokens from the reward function destabilizes training, resulting in purely textual responses and degraded performance.

## 4.5 ABLATION STUDIES

We examine variants in the architectural design and decoding strategies of **LVR** models.

**LVR with heads.** Analogous to the LM head in LLMs, which maps text hidden states to logits, we add a **LVR** head on top of the latent reasoning positions to transfer LLM hidden states into visual semantics. We evaluate two designs: (i) a 2-layer MLP without intermediate up-casting, and (ii) a gated linear unit (GLU) with an intermediate dimension expanded to $3\times$ the LM hidden size. The 7B results (Fig. 3) show that standard **LVR** consistently achieves the best performance across benchmarks. This is expected, as the **LVR** process is directly supervised in the joint semantic space of text and vision, ensuring no semantic gap between the LLM's last hidden states and those of **LVR**.

**Unrestricted LVR decoding.** As discussed in §3.3, we explored two alternative decoding strategies, **Mode Switching Loss** and **Latent End Token**, for variable-length **LVR** processes. However, **Mode Switching Loss** failed to encode stopping conditions, collapsing to zero **LVR** steps. The performance of **Latent End Token** is reported in Fig. 3, where we observe significant instability. Human evaluation indicates that this instability stems from unreliable distance measurements between **LVR** hidden states and the latent end token. Despite testing cosine similarity, L1, and L2 distances under varying thresholds, the model often failed to terminate properly and max out generation steps. We expect future work on improving stability for fully free-form LATENT VISUAL REASONING.

## 5 CONCLUSION

In this paper, we presented **LVR** as a novel multimodal reasoning paradigm that unifies latent reasoning over visual tokens with standard text generation. By extending the Vision–Projector–LLM structure and training with supervised fine-tuning plus GRPO-based reinforcement learning, **LVR** achieves stable hybrid reasoning. Experiments show substantial gains on perception-intensive benchmarks, demonstrating that reasoning jointly over latent visual and textual spaces offers a promising direction for future multimodal reasoning.

## 6 ETHICS STATEMENT

Technological innovations in multimodal large language models present both opportunities and risks. The impact of our proposed latent reasoning framework, associated decoding strategies, and evaluation benchmarks depends heavily on data quality and intended use. Ethical deployment requires that all training data be sourced responsibly and in compliance with legal and ethical standards, alongside safeguards for individual data rights. In the absence of comprehensive regulation, responsibility lies with practitioners to ensure proper use. Biases in either supervised finetuning data or reinforcement learning rewards may propagate disparities, particularly affecting underrepresented groups and undermining fairness and generalizability. To mitigate these risks, we emphasize adherence to ethical principles in system design, including transparency in training data and modeling choices, open-source releases to support accountability, and protective measures for vulnerable populations.

## 7 REPRODUCIBILITY STATEMENT

Our experiments utilized open-source models (Qwen-2.5-VL) and datasets (VISUAL COT, VIRL) for training. Training was conducted using open-source tools, including the Huggingface Trainer and DeepSpeed frameworks. To ensure full reproducibility, we will release our complete code base, model weights, and a corresponding Docker file, allowing for a complete replication of our setup.

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

# A APPENDIX

## A.1 USAGE OF LLMS

LLMs were used in this research project for coding assistance and writing support. Specifically, they were employed to generate helper functions, implement data loaders, proofread text, and suggest LaTeX formatting.

