**(a)  Question: What is the color of the cart?**

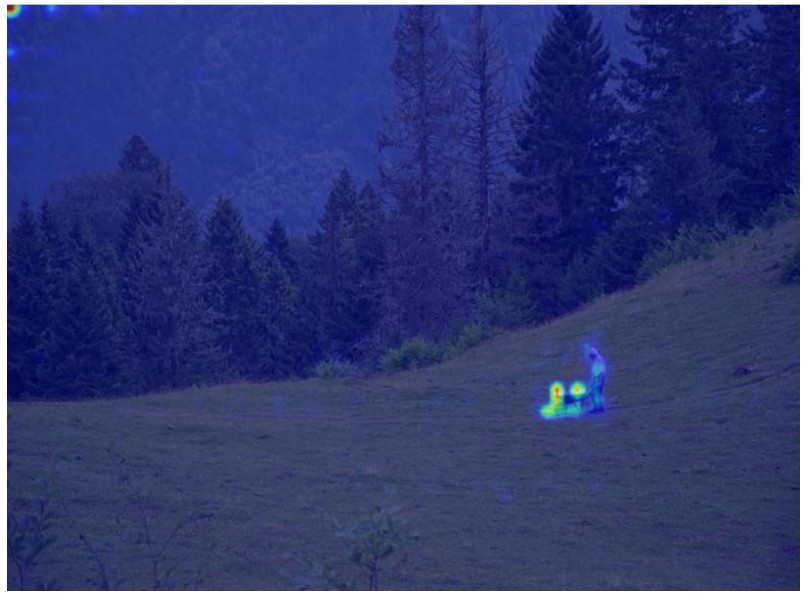 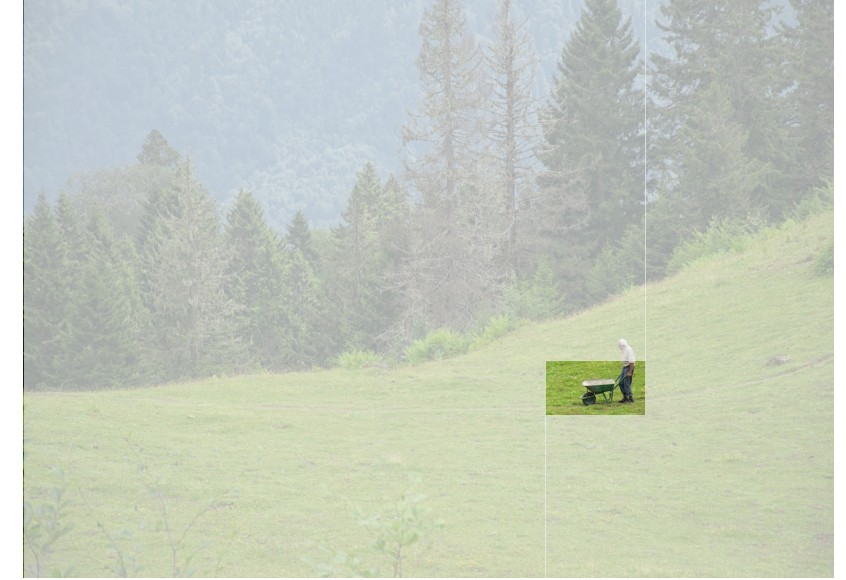 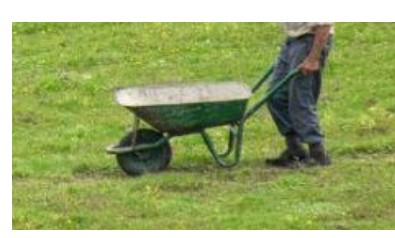

**(b) Question: What is the material of the glove?**

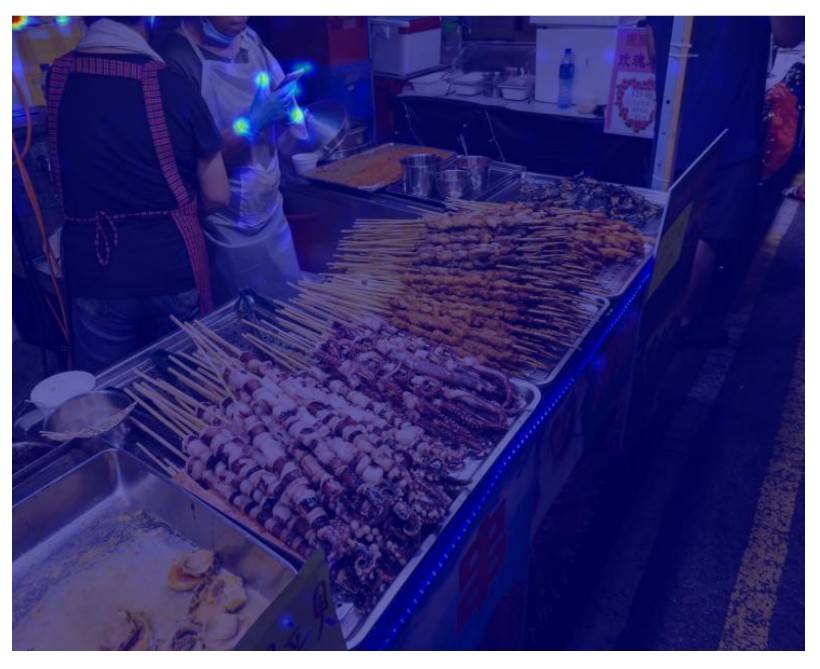 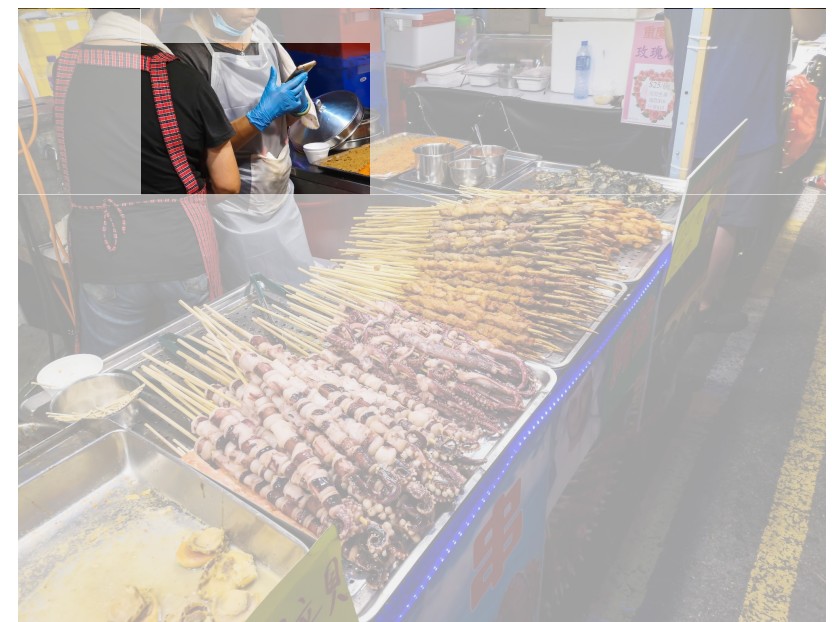 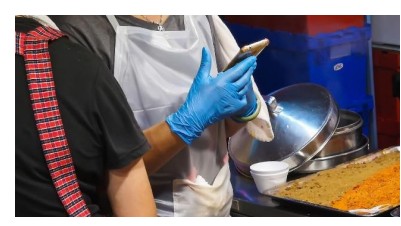

**(c) Question:  Is the kid with black shirt on the left or right side of the kid with blue shirt?**

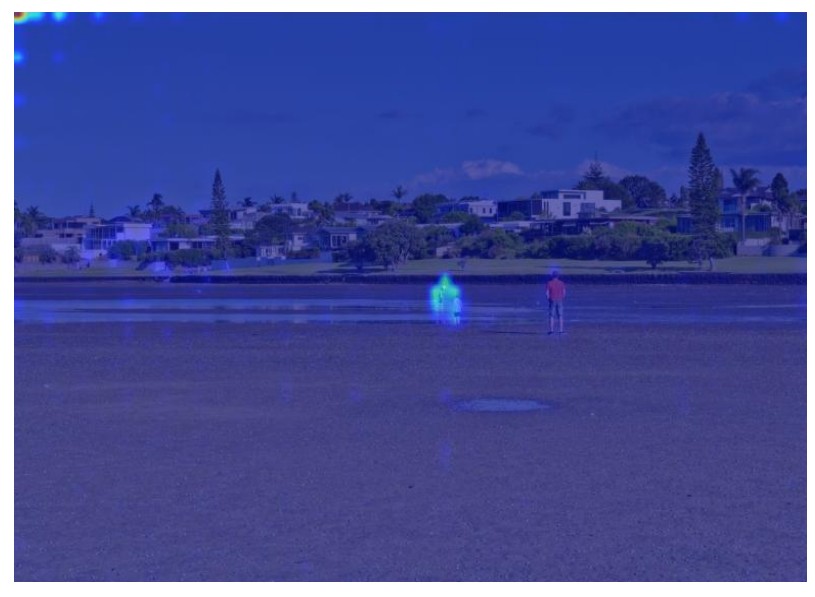 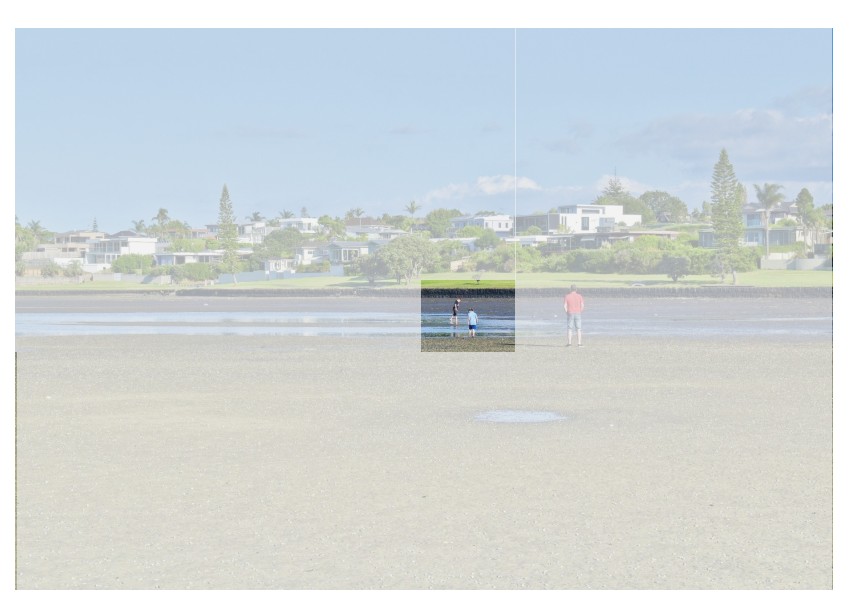 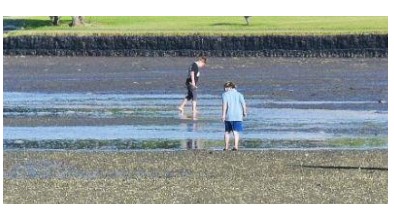

**LVR Attention Map on Image**          **Original Image (with Shades)**          **Zoom-In Area**