# OpenReview forum: "Latent Visual Reasoning"
_ICLR.cc/2026/Conference — ICLR 2026 Poster_

### Official Review · Reviewer_w9J6 · 2025-10-26

**Soundness:** 3
**Presentation:** 3
**Contribution:** 2
**Rating:** 4
**Confidence:** 3

**Summary:**

The paper presents Latent Visual Reasoning (LVR), a method that enables multimodal large language models to reason directly within their latent visual embedding space. LVR introduces a mechanism for switching between normal text generation and a special latent reasoning mode, indicated by specific tokens. In this mode, the model manipulates internal visual representations relevant to the question before producing text outputs. The training process has two stages: supervised fine-tuning using bounding-box annotations to align latent states with image regions, and reinforcement learning that refines reasoning quality.

Experiments are conducted on Qwen2.5-VL backbones at 3B and 7B scales. LVR shows the largest gains on fine-grained visual reasoning benchmarks such as MMVP, MMStar, and MM-Vet, compared to both “think-about” (text-only reasoning) and “think-with” (external visual reasoning) baselines. It performs particularly well on tasks requiring precise spatial, compositional, or attribute-based reasoning. The paper also introduces new decoding strategies for latent reasoning, comparing fixed-length and adaptive termination approaches. Training efficiency and decoding stability are analyzed through detailed ablations.

**Strengths:**

1. Simple modification of standard MLLM pipelines without external tools.
2. Adapts reinforcement learning to hidden-state spaces effectively.
3. Uses fixed encoders and adaptive batching for efficient multi-step reasoning and conceptually extends beyond CoT into “machine mental imagery.”
4. Open weights and code for reproducibility.

**Weaknesses:**

1. Gains are strongest on perception-heavy benchmarks, but less so on others (e.g., Relative Reflect).
Question: Have the authors tested multi-image or compositional reasoning tasks? Would LVR generalize beyond single-image settings?

2. The variable-length LVR decoding strategies (LatentEnd Token, Mode Switching Loss) perform inconsistently; only the fixed-token variant works reliably.
Question: Why does the model fail to self-terminate properly? Could an adaptive controller or entropy-based stopping improve this?

3. The paper compares against open models but omits newer ones like LLaVA-OneVision or InternVL 3.5, which are directly relevant.
Question: How does LVR compare to these recent omni-modal reasoning models?

4. Since LVR heavily relies on Visual-CoT training data, there’s risk it memorizes dataset patterns instead of general reasoning.
Question: Have they tested on out-of-distribution datasets to measure transfer robustness?

**Questions:**

See questions above

---

> ### Author Response · Authors · 2025-11-19
> **Re: Reviewer w9J6**
>
> Below is our response to the inspiring comments and suggestions.
>
> ---
>
> ### **W1.**
>
> We thank the reviewer for raising this question about generalization beyond single-image settings. The JigSaw benchmark serves as a multi-image understanding task for MLLMs: a region of the main image is masked out, and the model must select the most plausible candidate images from multiple visually similar options (e.g., portraits of public figures, buildings, crowds). This requires compositional analysis across images rather than single-image perception. As shown in Table 2, LVR models maintain strong performance on this task, suggesting that **the perceptual improvements learned from single-image training data generalize effectively to multi-image scenarios.**
>
> ---
>
> ### **W2.**
>
> We appreciate the reviewer’s insights in stopping mechanisms, which remain a core challenge in latent reasoning research. Our paper investigates two methods, **Latent End Token** and **Mode Switching Loss**, attempting to realize variable-length latent reasoning.
>
> **LatentEnd Token** is a learnable vector with the same dimensionality as the model’s hidden state. During SFT, this vector is inserted between the last visual token and \<lvr_end\>, and the model is trained to reproduce it immediately after the true final visual token, across varying LVR lengths. To avoid distorting the hidden-state space, we initialize this vector with magnitudes $k / \sqrt{d_{\text{model}}}$ for k $\in \{1, 2, 4\}.$
>
> At inference, we terminate LVR when the current latent state is sufficiently close to this learned vector (using Euclidean distance, cosine similarity, etc.). However, as shown in Table 3, even after extensive grid search, the signal remains unreliable: the model often triggers premature termination or, more critically, fails to terminate at all, causing significant degradation in downstream performance.
>
> **Mode Switching Loss.** serves as a surrogate confidence-based controller. At each latent step, we compute the next-token distribution using the lm_head (even though latent states do not map to meaningful text tokens). We then (1) minimize \<lvr_end\> predictions at intermediate latent steps, and (2) encourage the final step to predict \<lvr_end\>. This effectively treats the token distribution as a learned confidence measure for when to stop.
>
> In practice, however, the objective collapses: the model learns to predict \<lvr_end\> immediately after \<lvr_start\>, producing zero latent steps. We hypothesize that the lm_head discovers a degenerate shortcut that bypasses the intended mode-switching dynamics.
>
> We note that very recent work has proposed alternative strategies—for example, [1] introduces an accumulative exit probability that triggers termination once a threshold is reached. We believe continued exploration of such mechanisms will help the community converge toward more principled and reliable stopping criteria for latent reasoning.
>
> [1] https://arxiv.org/pdf/2510.25741v2
>
> ---
>
> ### **W3**
>
> | Model         | $V^{*}$ | MMVP | Counting | IQTest | JigSaw | RelativeReflect | SpatialRelation |
> |:-------------:|:-----:|:----:|:--------:|:------:|:------:|:----------------:|:----------------:|
> | LLaVa-OV-7B   | 74.4 | 76.0 | 71.7     | 28.0  | 54.0   | 29.9           | 79.0           |
> | InternVL-3.5  | 71.2  | 73.3 | 70.8     | 33.3  | 58.0   | 41.8            | 87.4           |
> | LVR-7B        | **81.2**  | 72.0 | 69.2     | 28.7   | 52.7   | **42.5**            | **89.5**            |
>
>
> We further evaluated two of the most recent strong baselines, LLaVA-OneVision-7B (Qwen2-based) and InternVL-3.5 (Qwen3-based). Notably, both models employ visual encoding modules that are architecturally different from ours, yet LVR still shows substantial advantages on benchmarks such as V-Star, Relative Reflect, and Spatial Reasoning.
>
> These results suggest that **latent visual reasoning provides gains that extend beyond what can be achieved through improved training recipes or stronger foundation LLMs**, particularly in fine-grained visual detail understanding and spatial reasoning. In fact, while MLLMs' performance has a strong correlation with its base model, **LVR serves as a universal paradigm that can be easily applied to other combinations of visual encoders and LLMs**. Consequently, we expect that applying the LVR mechanism on top of these evolving architectures will yield further improvements, highlighting its promise as a flexible and powerful paradigm for future multimodal reasoning research.

---

> > ### Author Response · Authors · 2025-11-27
> >
> > Dear Reviewer w9J6,
> >
> > Thank you again for your detailed and thoughtful review. We have posted our rebuttal and believe we have addressed your primary concerns. We would greatly appreciate your feedback on our response and are happy to answer any further questions or provide additional clarifications during the discussion period.
> >
> > Sincerely,
> >
> > The authors

---

> ### Author Response · Authors · 2025-11-19
> **Re: Reviewer w9J6 Continue**
>
> ---
>
> ### **W4**
>
> We would like to clarify that the Visual-CoT training data are sourced from the following datasets: COCO, GQA, OCR, TextVQA, VisualGenome, Visual7W, Flickr30k, DocVQA, InfographicsVQA, OpenImages, VSR, DUDE, SROIE, and CUB. **In contrast, the majority of our evaluation benchmarks are out of distribution relative to these sources.** For example, the $V^{*}$ benchmark is constructed from SA-1B, and the BLINK series (e.g., Counting, IQ-Test, etc.) consists of newly curated tasks collected from the Internet and annotated by experts.
>
> Importantly, **while most related works primarily evaluate models on in-domain training and testing distributions, our method demonstrates strong performance on out-of-distribution evaluation data.** This further highlights the novelty and contribution of our approach, showing that LVR generalizes beyond the domains it was trained on and remains effective under substantial distribution shifts.
>
> ---
>
> ### **Visualization**
>
> As commonly suggested to visualize the LVR process for a more intuitive understanding of how latent reasoning operates, we plotted the attention heatmaps associated with the generated LVR tokens on examples from the V-Star benchmark. **These visualizations are included as a PDF in the supplementary materials.**
>
> The attention weights are extracted from the middle layer of the LLM and averaged across all attention heads and all generated LVR tokens. As shown in the figures, the LVR tokens function as implicit latent states that consistently focus on query-relevant regions. **This provides concrete evidence that the latent visual reasoning process effectively aggregates the critical visual cues needed for downstream reasoning.**
>
> For instance, in the example question “What is the color of the cart?”, the LVR tokens correctly attend to the cart despite it being embedded within a large grassland. This supports the interpretability and validity of the proposed LVR paradigm.
> We will incorporate these visualizations into the appendix in the updated revision.

---

> > ### Comment · Reviewer_w9J6 · 2025-11-21
> >
> > Considering all of the above, I think the paper is good but not sufficient for publication in terms of scope and results, and I’m leaving the score as is.

---

> > > ### Author Response · Authors · 2025-11-21
> > > **Further discussion with Reviewer w9J6**
> > >
> > > We thank the reviewer for the follow-up. However, we kindly request further elaboration on the critiques regarding **"scope and results" as these specific concerns were not raised in previous discussions.** We are eager to address them and would like to offer the following clarifications:
> > >
> > > ### **Regarding Scope**
> > > We interpret this concern as referring to our research motivation. As detailed in the introduction, LVR is the first work to extend the MLLM reasoning space from text-only tokens to cross-modality. Our research vision is to enable AI systems to encode and decode all modalities within a universal hidden space, allowing for more comprehensive and robust reasoning trajectories. We view LVR as a pivotal step toward this paradigm shift.
> > >
> > >
> > > ### **Regarding Results**
> > > If this refers to model performance, we believe the evidence provided in the paper (and the additional results) is substantial. LVR has demonstrated effectiveness across a wide range of visual understanding tasks, including visual detail search, perception robustness, and spatial reasoning. Furthermore, as a general multimodal reasoning paradigm, LVR is architecture-agnostic and can be easily applied to various MLLMs.
> > >
> > >
> > > We believe the paper offers a sufficient contribution in both scope and empirical results for the domain and remain open to further discussion. **If there are specific flaws or missing experiments, please let us know so we can address them directly.**

---

### Official Review · Reviewer_eavF · 2025-10-27

**Soundness:** 3
**Presentation:** 4
**Contribution:** 3
**Rating:** 6
**Confidence:** 4

**Summary:**

The paper presents Latent Visual Reasoning (LVR), a novel method that allows multimodal models to reason directly with visual embeddings. Unlike existing models that reason with text or use external tools, LVR operates autoregressively within the visual space itself. The model learns to reconstruct important visual information related to a query by generating a sequence of latent states. This capability is developed in a two-stage training process, which starts with supervised fine-tuning and is later refined using reinforcement learning based on a modified GRPO algorithm. Experiments confirm the strength of this approach, showing it surpasses other methods on demanding visual perception benchmarks like MMVP and V*.

**Strengths:**

The primary strength of this paper lies in its introduction of Latent Visual Reasoning (LVR), a fundamentally new and well-motivated paradigm for multimodal reasoning. By enabling the model to reason directly within the visual embedding space, it provides an elegant alternative to purely text-based or tool-dependent approaches.  Furthermore, the methodology is well-conceived, featuring a logical two-stage training pipeline and a clever technical adaptation of reinforcement learning (GRPO_latent) to handle reasoning in a continuous latent space.

**Weaknesses:**

While the method excels at visual perception, its applicability to complex visual reasoning tasks remains unclear. For instance, in geometry problems where auxiliary lines are crucial, the practical value of the model's generated visual tokens is questionable and requires further validation. The work would be significantly strengthened by evaluations on more challenging reasoning benchmarks, such as MathVista, and by including more in-depth visual analyses that clearly illustrate the impact of these visual tokens on the reasoning process and its results.

**Questions:**

1. The current evaluation focuses almost exclusively on perception-based tasks. However, this leaves its performance on more complex visual reasoning tasks as an open question. Could the authors comment on the applicability of LVR to benchmarks like MathVista, which require more symbolic or multi-step reasoning? It is important to understand whether LVR's core mechanism of reconstructing visual tokens is sufficient for tasks that depend less on recognizing pixels and more on abstract logical deduction.

2. If the model focuses on an incorrect Region of Interest in an early step, does it have any mechanism to detect and correct this error in subsequent steps?

---

> ### Author Response · Authors · 2025-11-19
> **Re: Reviewer eavF**
>
> Below is our responses to the constructive comments and reviews:
>
> ---
>
> ### **W1.Q1.**
>
> We agree with the reviewer’s insightful comment regarding the need to evaluate LVR on more complex visual reasoning tasks. As suggested, we conducted additional experiments on MathVista using the lmms-eval evaluation suite. Specifically, we evaluated on the mathvista_testmini split and employed GPT-4o-mini as the LLM judge.
>
> Our LVR-7B model achieved 64.9% accuracy, compared to 60.0% for the vanilla Qwen-2.5-VL-7B baseline. **It is noteworthy that our training data does not explicitly involve math data. This improvement indicates that the latent visual reasoning process brings benefits beyond perception-oriented tasks**: by reconstructing and refining latent visual tokens, LVR helps distill key geometric and symbolic cues that support multi-step reasoning.
>
> These preliminary results suggest that LVR’s core mechanism extends naturally to more abstract reasoning scenarios. For future directions, we believe a more organic integration of LVR and COT could push the boundaries of multimodal reasoning.
>
> ---
>
> ### **Q2.**
>
> We appreciate the reviewer’s question regarding potential self-correction mechanisms. We first clarify that LVR does not explicitly predict or commit to a Region of Interest (ROI) during inference. Instead, the latent reasoning process produces a sequence of latent tokens that encode semantic information relevant to the query. As in standard text-based reasoning, an intermediate latent step may occasionally drift from the optimal trajectory. However, each subsequent latent token is generated conditionally and is again optimized to capture query-relevant semantics. This provides an implicit self-correction mechanism: later latent steps can refine or redirect earlier suboptimal states, analogous to how CoT reasoning improves itself through successive token predictions.
>
> ---
>
> ### **Visualization**
>
> As commonly suggested to visualize the LVR process for a more intuitive understanding of how latent reasoning operates, we plotted the attention heatmaps associated with the generated LVR tokens on examples from the $V^{*}$ benchmark. **These visualizations are included as a PDF in the supplementary materials.**
>
> The attention weights are extracted from the middle layer of the LLM and averaged across all attention heads and all generated LVR tokens. As shown in the figures, the LVR tokens function as implicit latent states that consistently focus on query-relevant regions. **This provides concrete evidence that the latent visual reasoning process effectively aggregates the critical visual cues needed for downstream reasoning.**
>
> For instance, in the example question “What is the color of the cart?”, the LVR tokens correctly attend to the cart despite it being embedded within a large grassland. This supports the interpretability and validity of the proposed LVR paradigm.
> We will incorporate these visualizations into the appendix in the updated revision.

---

> ### Comment · Reviewer_eavF · 2025-11-26
>
> Thank you so much for your visualization and response. They are very helpful and I will make my score keep positive.

---

### Official Review · Reviewer_9pGd · 2025-10-28

**Soundness:** 4
**Presentation:** 3
**Contribution:** 3
**Rating:** 6
**Confidence:** 5

**Summary:**

The paper introduces Latent Visual Reasoning (LVR) — a new paradigm for multimodal reasoning that allows autoregressive reasoning directly in the visual embedding (latent) space, rather than relying solely on textual Chain-of-Thought (CoT) reasoning or external visual tools. Experiments on visual reasoning and perception-intensive benchmarks (e.g., V*, MMVP, BLINK) show significant improvements over both “Think about Images” (text-only CoT) and “Think with Images” (external tool-based) baselines. LVR achieves up to 71.7% on MMVP, outperforming Qwen2.5-VL and PixelReasoner.

**Strengths:**

1. The concept of latent visual reasoning is novel and well-motivated.
2. The methodology is sound: clear architectural modifications, well-defined training objectives, and careful integration of latent-space reconstruction.
3. Experiments are comprehensive across multiple benchmarks and baselines (Qwen2.5-VL, PAPO, Vision-R1, PixelReasoner).
4. The work offers a scalable alternative to tool-based multimodal reasoning, reducing dependency on external APIs and handcrafted operations.

**Weaknesses:**

1. While conceptually strong, the paper provides little visualization or introspection of the latent reasoning process (e.g., what latent reconstructions “look like”). Future work could include latent trajectory visualizations or probing to show what is being “reasoned” in the latent space.
2. The reliance on a fixed latent reasoning length may restrict flexibility and generalization to variable reasoning complexity.

**Questions:**

1. Can you provide qualitative examples (e.g., t-SNE plots or attention maps) showing what information the latent visual reasoning reconstructs?
2. Could alternative stopping mechanisms (e.g., energy-based thresholds or learned confidence) improve variable-length reasoning stability?
3. Does LVR introduce latency at inference time due to latent-space propagation?

---

> ### Author Response · Authors · 2025-11-19
> **Re: Reviewer 9pGd**
>
> Below is our responses to the review, including visualization, stopping criteria and inferencing cost.
>
> ---
>
> ### **W1.Q1.**
>
> We appreciate the suggestion to visualize the LVR process for a more intuitive understanding of how latent reasoning operates. In response, we plotted the attention heatmaps associated with the generated LVR tokens on examples from the V-Star benchmark. **These visualizations are included as a PDF in the supplementary materials.**
>
> The attention weights are extracted from the middle layer of the LLM and averaged across all attention heads and all generated LVR tokens. As shown in the figures, the LVR tokens function as implicit latent states that consistently focus on query-relevant regions. **This provides concrete evidence that the latent visual reasoning process effectively aggregates the critical visual cues needed for downstream reasoning.**
>
> For instance, in the example question “What is the color of the cart?”, the LVR tokens correctly attend to the cart despite it being embedded within a large grassland. This supports the interpretability and validity of the proposed LVR paradigm.
>
> We will incorporate these visualizations into the appendix in the updated revision.
>
> ---
>
> ### **W2.Q2.**
>
> We appreciate the reviewer’s insightful question regarding alternative stopping mechanisms, which indeed remain a core challenge in latent reasoning research. Our paper explores two approaches, latent end token and mode switching loss, that align closely with the reviewer’s suggestions of energy-based thresholds and learned confidence.
>
> **Latent end token**.We introduce a learnable vector with the same dimensionality as the model’s hidden state. During SFT, this vector is inserted between the last visual token and \<lvr_end\>, and the model is trained to predict it immediately after the true final visual token across varying LVR lengths. To avoid perturbing the hidden-state space, we experiment with initializing this vector at magnitudes $k/\sqrt{d_\text{model}}$ for $k\in\{1,2,4\}$.
>
> At inference time, we terminate the LVR process when the current latent state is sufficiently close to this learned vector (euclidean, cosine, etc). As reported in Table 3, even after a grid search, we observe that the model struggles to learn a reliable signal: it often triggers early exits or, more problematically, never exits, the latter causing severe degradation in downstream performance.
>
> **Mode switching loss.** This method acts as a surrogate confidence mechanism. For each latent step, we compute the next-token distribution via the lm_head, even though latent states do not correspond to meaningful text tokens. We then (1) minimize the probability of \<lvr_end\> for all intermediate latent steps, and (2) encourage the final latent step to predict \<lvr_end\>. This repurposes the token distribution as a confidence score for deciding when to stop.
>
> However, in practice this objective collapses: the model immediately outputs \<lvr_end\> after \<lvr_start\>, yielding zero latent steps. We hypothesize that the lm_head discovers a shortcut by output \<lvr_end\> right after \<lvr_start\>, bypassing the intended mode-switching behavior.
>
> We note that very recent work has proposed additional strategies. For example, [1] introduces an accumulative exit probability that triggers termination once a threshold is reached. With continued efforts in this direction, we expect the community to converge toward more principled and robust approximations of optimal stopping criteria for latent reasoning.
>
> [1] https://arxiv.org/pdf/2510.25741v2
>
> ---
>
> ### **Q3.**
>
> **The inference cost for LVR is identical to that of regular MLLMs.** Our decoding procedure is implemented by minimally modifying the standard LLM decoding loop. In a conventional MLLM, each step takes the next-token distribution, applies a sampling strategy, and feeds the selected token into the next forward pass. In our LVR decoding, we simply maintain a flag indicating whether the model is in LVR mode. When generating text, the behavior is exactly the same as a standard decoder. When in LVR mode, instead of sampling from a token distribution, we take the last hidden state and feed it directly into the next step as the token embedding. This avoids any additional computation and there is no extra processing or overhead introduced by LVR during inference.

---

> > ### Comment · Reviewer_9pGd · 2025-11-20
> >
> > Thank you for your response. It addressed most of my concerns, and I have raised the score.

---

### Official Review · Reviewer_Zdot · 2025-10-30

**Soundness:** 2
**Presentation:** 3
**Contribution:** 3
**Rating:** 6
**Confidence:** 3

**Summary:**

This paper proposes latent visual reasoning for MLLMs, which supervises an autoregressive process on latent states with reconstruction of ground-truth visual semantic embeddings, resulting better results than previous "think about images" and "think with images" methods. The training stages include SFT with next token prediction combined with reconstruction, and a GRPO stage compatible with the generated latent sequences. The paper further investigates different decoding strategies and concludes that the method works best with a fixed number of reasoning steps.

**Strengths:**

1. The framework is novel by directly using MSE on hidden states and incorporating GRPO with modifications.
2. The paper is well-organized and clearly presented.
3. The experimental results are achieved by carefully controlling the conditions, making the performance advantage convincing.

**Weaknesses:**

Although the training stages incorporates RL, it can not avoid using critical region annotation, as GRPO yields only marginal improvement, while the MSE supervision on latents brings clear advantage over vanilla SFT. The latents are supervised only by the region semantics from the input image, which seems more like learning a visual grounding. Thus, the novelty advantage might be limited compared to other works proposing latent visual reasoning.

**Questions:**

1. How can the RL process improve the latent generation process? As I see in (5), the latents seem to be used as inputs with no gradients.
2. I find the "LVR with heads" confusing in the ablation section. Is it just some additional mapping parameters? Why is the performance worse with more parameters involved?

---

> ### Author Response · Authors · 2025-11-19
> **Re: Reviewer Zdot**
>
> Below is our responses to the constructive comments and questions from the reviewer.
>
> ---
>
> ### **W1**.
> We thank the reviewer for the thoughtful comment.We want to emphasize that the novelty of LVR lies in its contribution to the multimodal reasoning paradigm, rather than in learning a new visual grounding signal. **The motivation behind LVR is to expand the model’s reasoning search space from “selecting the best text token from a vocabulary” to "selecting the most useful information across both modalities."**
>
> Existing visual-grounding approaches operate by predicting bounding boxes and then relying on external tools for cropping and re-encoding before reasoning. In contrast, LVR performs the entire reasoning procedure internally within the model, without relying on external perception modules. We view this as an important step toward the long-term goal of unified multimodal models that reason over integrated semantic spaces.
>
> Regarding the reviewer’s reference to “other works proposing latent visual reasoning,” we would appreciate specific citations. To the best of our knowledge, as discussed in our Related Work, the closest works are [1, 2]. (We welcome further discussions if the reviewer is not referring to these two methods.) Both share a common intuition: they introduce a small number of learned latent tokens intended to summarize auxiliary images for downstream reasoning. However, LVR differs from them in two key aspects:
>
> a) **Granularity.**
>
> **Prior methods enforce strict constraints on the latent tokens.** For example, [1] uses a predefined, task-specific codebook, and [2] limits the model to exactly four latent tokens by pooling each auxiliary image into this fixed-size representation. These designs reflect their focus on generation of external signals or representations. In contrast, LVR imposes no such restrictions during training because our focus is token-level multimodal reasoning. LVR operates on fine-grained latent interactions across modalities rather than compressing images into fixed-size latent embeddings.
>
> b) **Generalizability.**
>
> **Both [1] and [2] require task-specific in-domain training, and their generalization beyond those domains remains unclear.** Based on their experimental results, it is unknown if these method training will generalize to Out-of-Distribution evaluation data. In contrast, LVR is trained on Visual-CoT supervision but evaluated on a diverse set of VQA benchmarks that are out-of-distribution, demonstrating strong generalization and robustness across tasks.
>
> We hope this clarification helps highlight why LVR is a meaningful advancement toward unified multimodal latent reasoning.
>
> [1] https://arxiv.org/pdf/2412.03548
>
> [2] https://arxiv.org/pdf/2506.17218
>
> ---
>
> ### **Q1**.
> We agree that the RL objective does not directly backpropagate gradients through the latent states, since the policy gradient is computed over the text-token distribution. However, as shown in (5), the downstream text generation is conditioned on the latent sequence, and this subsequent generation path does receive policy gradients. Through these gradients, the attention and FFN weights that mediate the influence of the latent states are updated. As a result, the latent reasoning steps receive indirect but meaningful optimization pressure: changes that make the latent sequence more conducive to producing high-reward text will be reinforced.
> **This design intentionally removes the reconstruction constraints**, which may introduce heuristic bias, allowing the model to adapt its latent reasoning process through outcome-based reinforcement learning, **which encourages more flexible latent trajectories**.
>
> ---
>
> ### **Q2**.
> The lvr_head is designed as the counterpart of the lm_head: just as the lm_head projects hidden states into the vocabulary distribution, the lvr_head projects LLM hidden states into the encoded visual semantic space. **Our initial hypothesis was that introducing this projection head might soften the reconstruction signal for the LLM**.
>
> However, in practice, variants of lvr_head underperform compared to the headless LVR design. This outcome is reasonable given that the reconstruction targets for LVR are the visual tokens after the multimodal projector, which not only aligns embedding dimensionality but also maps visual information into the LLM’s semantic space. **Therefore, the target lies in essentially the same representational space as the LLM hidden states. Adding an additional projection layer between them introduces unnecessary parameters and noise, which disrupts this alignment and ultimately degrades performance.**

---

### Official Review · Reviewer_T9HU · 2025-11-01

**Soundness:** 3
**Presentation:** 3
**Contribution:** 2
**Rating:** 4
**Confidence:** 3

**Summary:**

The paper introduces Latent Visual Reasoning (LVR), a novel paradigm that enables multimodal large language models to perform reasoning directly within the visual embedding space, rather than relying solely on textual chain-of-thought reasoning. LVR allows the model to interleave latent visual reasoning with text generation, jointly processing both modalities in a shared semantic space. The authors propose a two-stage training pipeline combining supervised finetuning for visual reconstruction and reinforcement learning with an adapted GRPO algorithm to refine reasoning dynamics. Experimental results show that LVR significantly improves fine-grained visual perception, spatial understanding, and robustness on benchmarks such as MMVP and V*, outperforming previous “Think about” and “Think with” image approaches.

**Strengths:**

1. The paper is very well-written and logically structured, making the core ideas easy to understand. The figures are clear and effectively illustrate the main concepts.

2. The motivation is straightforward and reasonable.

**Weaknesses:**

1. The paper mostly extends the idea of latent reasoning from text-only LLMs to multimodal models. While this is a reasonable step forward, it feels more like a technical adaptation than a fundamentally new idea. The authors could do a better job explaining why reasoning in the visual latent space offers a distinct advantage rather than just being a straightforward extension of existing methods.

2. The approach relies heavily on Visual CoT data and also requires identifying regions of interest (ROIs) for training, which makes it conceptually close to Visual CoT itself. However, there’s no head-to-head comparison with Visual CoT models in terms of both performance and inference cost. Without this, it’s hard to judge how much the proposed method actually improves over existing baselines.

3. Because the reasoning happens in the latent space, it’s not clear what the model is actually doing during the latent visual reasoning phase. The paper doesn’t provide any qualitative examples or visualizations of these hidden processes, which makes the claimed reasoning mechanism somewhat opaque.

**Questions:**

N/A

---

> ### Author Response · Authors · 2025-11-19
> **Re: Reviewer T9HU**
>
> Bellow is our response to the thoughtful comments from the reviewer:
>
>
> ---
>
> ### **W1.**
> We appreciate the reviewer’s comments regarding the motivation behind latent visual reasoning. We would like to clarify that our proposed LVR paradigm is not a straightforward extension of textual latent reasoning, but is motivated by a fundamentally different challenge in multimodal learning. **To summarize, we differ in motivation, learning objective and training methods.**
>
> First, the objective of the pioneer textual latent reasoning work,  Coconut [1], is to perform continuous CoT as a form of latent `"Breadth First Search" , effectively compressing multiple textual reasoning traces into a fixed number of steps. In contrast, LVR is designed to address the modality gap that emerges during multimodal reasoning. Rather than compressing reasoning trajectories, LVR expands the reasoning search space across modalities, enabling the model to actively integrate visual semantic information during intermediate reasoning steps.
>
> Second, because the goals differ, the training strategies also diverge. Coconut employs a “masked reasoning step’’ objective, gradually removing past textual reasoning and encouraging the latent space to conclude past textual steps, with no explicit supervision signal tied to external semantics. LVR, however, is explicitly supervised to recover query-relevant visual semantics during the latent reasoning process. This changes the nature of the latent reasoning task: LVR is not latent for efficiency or compression of complexity, but latent for cross-modal reasoning.
>
> Given these differences in motivation, objective, and methodology, **we position LVR as an innovative multimodal reasoning paradigm rather than a technical adaptation of textual latent reasoning.** We appreciate the reviewer’s suggestion and will strengthen the related work section to clarify these distinctions.
>
> ---
> ### **W2.**
>
> We appreciate the reviewer’s comment on the ``head-to-head’’ comparison.
>
> First, regarding the comparison with Visual-CoT, the original Visual-CoT model was built on LLaVA-1.5, which was trained with substantially less data and compute than Qwen2.5-VL, the foundation model used by LVR and all baselines in our paper. **To ensure a fair ablation, we therefore retrained Qwen2.5-VL on the Visual-CoT dataset and reported its performance in Table 1 under “SFT.”** These results show that directly integrating latent visual semantics into the reasoning process enables the model to more effectively exploit ROI signals and consistently improves VQA performance over the SFT baseline.
> We also include PixelReasoner as a stronger and more representative comparison. PixelReasoner belongs to the “thinking with images” family of methods that perform explicit query-driven image editing (e.g., cropping subregions) as intermediate reasoning steps. It is therefore well aligned with the subimage-enhanced reasoning paradigm that Visual-CoT follows, and it uses the same underlying foundation model as LVR. Our experiments demonstrate that LVR offers more reliable gains on fine-grained visual understanding tasks, highlighting that reasoning directly in the latent visual space provides advantages beyond the explicit image-editing pipeline used in PixelReasoner and Visual-CoT.
>
> ---
> ### **W3.**
>
> We appreciate the suggestion to visualize the LVR process for a more intuitive understanding of how latent reasoning operates. In response, we plotted the attention heatmaps associated with the generated LVR tokens on examples from the $V^{*}$ benchmark. These visualizations are included as a PDF in the supplementary materials.
> The attention weights are extracted from the middle layer of the LLM and averaged across all attention heads and all generated LVR tokens. As shown in the figures, the LVR tokens function as implicit latent states that consistently focus on query-relevant regions. **This provides concrete evidence that the latent visual reasoning process effectively aggregates the critical visual cues needed for downstream reasoning.**
> For instance, in the example question “What is the color of the cart?”, the LVR tokens correctly attend to the cart despite it being embedded within a large grassland. This supports the interpretability and validity of the proposed LVR paradigm.
> We will incorporate these visualizations into the appendix in the updated revision.

---

> ### Author Response · Authors · 2025-11-27
>
> Dear Reviewer T9HU,
>
> Thank you again for your detailed and thoughtful review. We have posted our rebuttal and believe we have addressed your primary concerns. We would greatly appreciate your feedback on our response and are happy to answer any further questions or provide additional clarifications during the discussion period.
>
> Sincerely,
>
> The authors

---

> > ### Comment · Reviewer_T9HU · 2025-11-28
> >
> > Thanks for the response. The rebuttal addresses my concerns, and I raise the score.

---

> ### Author Response · Authors · 2025-11-28
>
> Hi Reviewer T9HU,
>
> Thank you very much for your follow-up and for considering raising the score. We want to let you know that the review system is currently undergoing technical adjustments, and the scores are temporarily not editable. We will continue to monitor the system and will keep you updated as soon as the editting becomes available.
>
> We truly appreciate your time, effort, and thoughtful engagement with our work.
>
> Best regards,
> Authors

---

### Meta-Review · Area_Chair_TJad · 2026-01-03

**Summary:**

Although multimodal large language models (MLLMs) have advanced through CoT reasoning, reasoning remains primarily confined to the linguistic domain. Visual information is often treated statically. This paper introduces a new approach called Latent Visual Reasoning (LVR), which performs autoregressive reasoning directly within the visual embedding space. The model projects images onto visual tokens in a shared semantic space and learns to generate latent states that reconstruct the visual tokens necessary for the response. Alternating between LVR and text generation improves performance on perceptually-aggregated VQA tasks, and reinforcement learning using GRPO optimizes the balance between the two.

The review highlighted the following six main points: (1) novelty and positioning, (2) insufficient baseline comparisons, including those with Visual CoT and the latest VLM, (3) a lack of interpretability and visualization, (4) the limited contribution of RL/GRPO, (5) concerns about generalization and applicability, and (6) the instability of variable-length LVR. In response, the authors clarified the method's positioning and differences, the intent of the evaluation setup, the role of the RL stage, and the rationale behind the current state and design of variable-length decoding. The AC judges that the authors have generally addressed these concerns appropriately.

Therefore, this paper is expected to contribute to the community by proposing a new reasoning paradigm, and the AC recommends acceptance.

**Reviewer Concerns:**

The reviewers raised the following concerns:
- Novelty and positioning
- Lack of baseline comparisons (especially Visual CoT and the latest VLM)
- Lack of interpretability and visualization
- Small contribution of RL/GRPO
- Concerns regarding generalization and applicability
- Unstable variable-length LVR

The AC considers these concerns resolved by the reviewers' responses.

During the discussion process, reviewer w9J6 raised concerns about the scope and results of this paper. However, the AC considers this comment to lack specificity, making constructive discussion difficult.

**Reviewer Scores:**

Reviewer T9HU has a score of 4 and says they would raise it because they are satisfied with the authors' responses.

Reviewer Zdot has a score of 6 and did not participate in the final discussion. This reviewer's main concern is RL/GRPO's small contribution. Since the authors responded appropriately to this concern, the AC believes that the score will not drop significantly from a 6.

Reviewer 9pGd gave a score of 6 and commented that they raised the score because they were satisfied with the authors' responses.

Reviewer eavF gave a score of 6 and stated that they maintained the positive score because they were satisfied with the authors' responses.

Reviewer w9J6 gave a score of 4 and stated that they would maintain this score following the discussion. The remaining concerns are about scope and results. The authors addressed these concerns, but this reviewer did not comment. The AC considers it unlikely that this reviewer will raise their score.

---

### Decision · Program_Chairs · 2026-01-26

Accept (Poster)